# Antispasmodic Effect of Bergamot Essential Oil on Rat Isolated Gut Tissues

**DOI:** 10.3390/pharmaceutics14040775

**Published:** 2022-04-02

**Authors:** Laura Rombolà, Marilisa Straface, Damiana Scuteri, Tsukasa Sakurada, Shinobu Sakurada, Maria Tiziana Corasaniti, Giacinto Bagetta, Luigi Antonio Morrone

**Affiliations:** 1Department of Pharmacy, Health and Nutritional Sciences, Section of Preclinical and Translational Pharmacology, University of Calabria, 87036 Rende, Italy; laura.rombola@unical.it (L.R.); marilisastraface@gmail.com (M.S.); g.bagetta@unical.it (G.B.); 2Regional Center for Serious Brain Injuries, S. Anna Institute, 88900 Crotone, Italy; 3First Department of Pharmacology, Daiichi College of Pharmaceutical Sciences, Fukuoka 815-8511, Japan; tsukasa@daiichi-cps.ac.jp; 4Department of Physiology and Anatomy, Tohoku Pharmaceutical University, Sendai 981-8558, Japan; sakura@tohoku-pharm.ac.jp; 5Department of Health Sciences, University “Magna Graecia” of Catanzaro, 88100 Catanzaro, Italy; mtcorasa@unicz.it

**Keywords:** bergamot essential oil, limonene, α-pinene, linalyl-acetate, linalool, enteric tissues, organ bath

## Abstract

Preclinical data indicate that bergamot essential oil (BEO) can modulate the synaptic functions within the central nervous system (CNS). Particularly, several data shows that essential oil is endowed with reproducible analgesic and anxiolytic effects that may derived from the ability to modulate the excitatory and inhibitory neurotransmission in the CNS. Although there are differences in the functional complexity of the enteric nervous system (ENS), it is likely that the phytocomplex has biological properties in gut superimposable to those showed in the CNS. Accordingly, the aim of this study was to investigate ex-vivo the effect of bergamot essential oil and its main constituents on the contractile activity of rat isolated colon, jejunum and ileum induced by different muscle stimulants such as acetylcholine (10^−6^ M) and potassium chloride (80 mM). Our present data demonstrate that BEO inhibits cholinergically- and non cholinergically-mediated contractions in rat isolated gut and that linalool is the most active component. These results suggest that the phytocomplex might be useful in the treatment of spastic disorders in ENS mainly characterized by the presence of pain; incidentally, irritable bowel syndrome (IBS) is a painful condition in which a role for neurotransmitter dysfunction has been envisaged. More investigation is required for clinical translation of the present data.

## 1. Introduction

Bergamot (*Citrus bergamia* Risso & et Poiteau, syn.: *Citrus aurantium* L. subsp. *bergamia* (Risso et Poiteau), an aromatic citrus fruit belonging to genus Citrus, family Rutaceae, is worldwide used to extract the essential oil from the peel by a cold-pressing procedure [1]. Bergamot essential oil (BEO) includes monoterpene and sesquiterpene hydrocarbons and their oxygenated derivatives in the volatile fraction and coumarins and psoralens in the non-volatile fraction [2,3]. In traditional and folk medicine BEO was used as antiseptic and anthelmintic while in aromatherapy it is used to improve mood, depression, anxiety [4] and behavioral and psychological symptoms of dementia [5]. Over the last decade, preclinical results from our laboratories, elucidated cellular mechanisms underlying rational basis for the therapeutic use of BEO. Particularly, the phytocomplex can produce neurobiological effects in animal models, which originated, partially, from modulation of fine mechanisms involved in synaptic plasticity [6,7,8,9] in the central nervous system (CNS). Briefly, systemic administration of BEO stimulates the release of discrete excitatory and inhibitory amino acids, as glutamate and γ-aminobutiric acid (GABA), respectively, in the hippocampus of rat. Particularly, the use of superfused hippocampal synaptosomes allowed to discriminate a Ca^2+^-dependent mechanism, when low concentrations of BEO were used, and a non-Ca^2+^-dependent carrier mechanism at high concentrations of phytocomplex [6]. An interference of the essential oil with different neurotransmissions was also shown in preclinical behavioural studies. In fact, systemic administration of GABA_A_ [10] or 5-HT-1 [11] receptor antagonists, partially, reverted the anxiolytic-like/relaxant effects induced by BEO in different tasks in rodents [9], suggesting that it may modulate different pathways. Interestingly, BEO shows analgesic activity in acute nociceptive [12,13,14,15] and neuropathic pain models [16,17] originated, at least in part, through the involvement of opioid system [14,16,17].

In mammals, the enteric nervous system (ENS) and the CNS both contain integrative neural circuitry and similarities between them [18]. It is known that ENS regulates multitasking functions by a unique autonomic network of neurons that operates independently although it communicates with the CNS. For these abilities ENS is considered as a “second brain” [19]. Despite the diverse functional complexity of ENS, it is likely that essential oil may show in the gut biological properties superimposable to those observed in the CNS.

In this study, using isolated organ bath technique, the effects of BEO were characterized on cholinergically- and non-cholinergically-mediated contractions induced respectively by acetylcholine (ACh) and potassium chloride (KCl) on different tissues (colon, jejunum and ileum) of the enteric system of rat. It was also investigated the activity of single oxygenated components of the volatile fraction of BEO such as limonene, α-pinene, linalyl-acetate and linalool reported to contribute in the pharmacological properties of the essential oil [13,15,20,21].

## 2. Materials and Methods

### 2.1. Materials

Atropine (ATR), ACh, (R)-(+)-limonene, α-pinene, linalyl acetate and (-)-linalool were purchased from Sigma-Aldrich (Milan, Italy). The crude bergamot essential oil was kindly provided by “Capua Company1880 S.r.l.”, Campo Calabro (Reggio Calabria, Italy). Chromatographic analysis of BEO reports: (R)-(+)-limonene ~48.6%, α-pinene ~3.2%, linalyl acetate ~23.6%, (-)-linalool ~5.5% of the total volume (*v*/*v*). ACh and ATR (Sigma-Aldrich, Milan, Italy) were dissolved in distilled water. BEO and its constituents were dissolved in dimethylsulphoxide (DMSO) (Sigma-Aldrich, Milan, Italy) and its amount was less than 1% of the tissue bath volume.

### 2.2. Animals

The animal study protocol was approved by Italian Ministry of Health (Rome, Italy); (protocol code 700A2N.6TI, date of approval: 13 March 2018). Animal procedures were conducted under approved guidelines by the University Animal Welfare Committee (OPBA) of the Department of Pharmacy, Health and Nutritional Sciences of University of Calabria (Italy) (date of approval: 7 March 2018) and in accordance with the Council Directive (86/609/EEC) and D.Lgs 26/2014 in order to apply the principles of reduction and refinement. Experiments were performed on adult male rats (250–300 g), obtained from Charles River Laboratories (Calco-Lecco, Italy). Animals were housed in 36 × 18.5 × 24 cm clear polyethylene cages in a room with controlled temperature (22 ± 1 °C) and under a light-dark schedule (lights on 7 a.m. to 7 p.m.) for at least 7 days before being used. Food and water were freely available. The rats were sacrificed by exposure to 4% isoflurane air followed by cervical dislocation.

### 2.3. Experimental Protocol

Following the opening of the abdominal cavity, whole segments of jejunum, ileum and colon were excised and the content gently flushed out with Krebs solution consisting of (mM): NaCI 119; KCI 4.5; MgS0_4_ 2.5; NaHCO_3_ 25; KH_2_P0_4_ 1.2; CaCl_2_ 2.5; glucose 11.1 (Sigma-Aldrich, Milan, Italy). The segments, cleaned from adjacent and connective tissue were cut into full thickness strips along the longitudinal (15 mm long and 5 mm wide) and were vertically suspended in two channel organ bath containing 40 mL of oxygenated (95% O_2_ and 5% CO_2_) Krebs solution at 37 °C. Before use, the preparations were allowed to equilibrate for 45 min. Then, contractions were evoked by exogenously applied ACh (10^−6^ M) and KCl (80 mM); the latter substances were administrated and left in the tissue baths for 5 min to obtain reproducible responses to use as control. After obtaining of three reproducible contractions, BEO (2.5 × 10^−5^ to 2.5 × 10^−3^% *v*/*v*), linalool (2.5 × 10^−6^ M to 2.5 × 10^−4^ M), limonene (2.5 × 10^−6^ M to 2.5 × 10^−3^ M), linalyl acetate (2.5 × 10^−6^ M to 2.5 × 10^−3^ M) and α-pinene (2.5 × 10^−6^ M to 2.5 × 10^−3^ M) (Sigma-Aldrich, Milan, Italy) were added to the bath 5 min before ACh. The distal end of each segment was tied to an organ holder and the proximal end was secured with a silk thread to an isometric force transducer (Ugo Basile 7005, Gemonio, Varese, Italy). Mechanical activity was digitized on an A/D converter, visualized, recorded and analyzed on a personal computer using the Data Capsule-Evo system (Ugo Basile, Gemonio, Varese, Italy).

### 2.4. Statistics

Data are expressed as the mean ± S.E.M, *n* = 3–6 in all experiments. GraphPad PRISM 7.0 for Windows (Graph-Pad Software, La Jolla, CA, USA) was used for statistical analysis and graphics production. Effects were considered statistically significant if they had a null hypothesis probability lower than 5% (*p* ˂ 0.05). Statistical significance of any difference between unpaired data was determined using Student’s test (*t*-test).

## 3. Results

### 3.1. Effect of BEO on Acetylcholine-Evoked Contraction

Administration of ACh (10^−6^ M) elicited a contraction (indicated as 100%) in rat isolated colon, jejunum and ileum that was completely inhibited by pre-treatment with ATR (10^−6^ M) for 5 min (Figure 1) highlighting the involvement of muscarinic receptors.

Administration of BEO to the bath produced a concentration-dependent decrease of the cholinergic contraction in all gut tissues investigated (Figure 1). Particularly, in colon a reduction to 92.2 ± 4.1%, 70.5 ± 3.3% and 5.0 ± 5.0% was measured at the concentration of 2.5 × 10^−5^%, (*v*/*v*), 2.5 × 10^−4^%, (*v*/*v*) and 2.5 × 10^−3^%, (*v*/*v*), respectively. In jejunum, values of 96.4 ± 3.6%, 60.2 ± 5.5% and 1.0 ± 1.0% were measured with increased volumes of phytocomplex. Highest effects were observed in ileum where cholinergic contraction was reduced by BEO to 83.2 ± 6.2%, 58.9 ± 6.5% and 0.75 ± 0.7%, respectively (Figure 1).

### 3.2. Effect of BEO Constituents on Acetylcholine-Evoked Contraction

In colon, concentration of 2.5 × 10^−3^ M of limonene, α-pinene and linalyl acetate reduced the contraction elicited by ACh (10^−6^ M) to 74.2 ± 13.2%, 93.3 ± 3.5% and 93.5 ± 1%, respectively. Lower concentrations of the compounds failed to affect muscarinic contraction (Figure 2). Interestingly, concentrations of 2.5 × 10^−5^ M and 2.5 × 10^−4^ M of linalool reduced in a concentration-dependent manner muscarinic contraction to 77.28 ± 1.53% and 2.5 ± 2.1%, respectively. Linalool did not show effect on muscarinic contraction at concentrations of 2.5 × 10^−6^ M.

In jejunum, limonene did not affect cholinergic contraction, while α-pinene was able to induce a statistical significant inhibitory effect (92.3 ± 3.5%) only at the highest concentration (2.5 × 10^−3^ M). On the other hand, linalyl acetate induced a concentration-dependent (2.5 × 10^−5^–2.5 × 10^−3^ M) decrease of ACh contraction to 94.6 ± 2.9%, 81.8 ± 9.1% and 5.0 ± 2.9% respectively. Similarly, linalool reduced, in a concentration-dependent manner (2.5 × 10^−6^–2.5 × 10^−4^ M), the contraction induced by ACh to 93.5 ± 5.9%, 67.7 ± 6.4% and 0.17 ± 0.17%, respectively (Figure 2).

In ileum, limonene and α-pinene were able to elicit a statistical significant inhibitory effect on muscarinic contraction only at the highest concentration used (2.5 × 10^−3^ M) reaching the values of 93.5 ± 3.4% and 92.8 ± 3.6%, respectively. Conversely, linalyl acetate (2.5 × 10^−5^–2.5 × 10^−3^ M) and linalool (2.5 × 10^−6^–2.5 × 10^−4^ M) induced a concentration-dependent decrease of cholinergic contraction. Particularly, linalyl acetate induced a reduction to 88.4 ± 3.2%, 75.4 ± 6.4% and 67.7 ± 8.0% and linalool to 84.7 ± 2.3%, 52.1 ± 3.4% and 0.6 ± 0.6%, respectively (Figure 2). The spasmolytic effects of linalool were obtained at concentrations ten times less than linalyl acetate.

### 3.3. Effect of BEO and Linalool on KCl-Evoked Contraction

Administration of KCl (80 mM) to the bath elicited a contraction (indicated as 100%) in rat isolated colon, jejunum and ileum that was not modified by pre-treatment with ATR (10^−6^ M) for 5 min suggesting that muscarinic receptors were not involved (Figure 3).

Overall, increased concentration of BEO (2.5 × 10^−5^–2.5 × 10^−3^%, (*v*/*v*)) induced a concentration-dependent decrease of KCl contraction and, similarly to what was observed with cholinergic contractions, its most active constituent was linalool. For the sake of brevity, we only report the effect of linalool (Figure 3).

Particularly, in colon, the phytocomplex induced a statistically significant decrease of KCl contraction to 75.67 ± 3.5% and 11.33 ± 6.6% with the volumes of 2.5 × 10^−4^–2.5 × 10^−3^%, (*v*/*v*) while linalool induced a reduction to 64.3 ± 8.5% and 2.94 ± 2.8% at the concentrations of 2.5 × 10^−5^–2.5 × 10^−4^ M, respectively (Figure 3). In jejunum, the essential oil induced a statistically significant reduction to 74.6 ± 2.4% and 24.83 ± 8.4% with the medium or high volume used. Linalool reached a statistically significant inhibitory effect (19.84 ± 13.6%) only at the concentration of 2.5 × 10^−4^ M (Figure 3).

BEO elicited similar results also in ileum and decreased KCl contraction to 54.4 ± 7.9% or 13.8 ± 8.6% at medium and high volumes used. In ileum, linalool induced a statistically significant reduction of KCl contraction at medium and high concentrations used (65.6 ± 9.4% or 15.0 ± 2.2%, respectively) (Figure 3).

## 4. Discussion

The results obtained show that contractions induced by ACh in colon, jejunum or ileum of rat are inhibited by BEO in a concentration-dependent manner and abolished by the highest volume tested. Similarly, BEO decreased KCl contractions in gut tissues, suggesting that phytocomplex counteracts both cholinergically- and non cholinergically-mediated contractions. The ability of BEO to inhibit non cholinergically-mediated contractions in isolated gut supports recent data by Straface and colleagues (2020) [21] and is further confirmed by some preliminary results obtained on atropine-insensitive contractions evoked by substance P in rat intestine (Morrone et al., unpublished observations).

Incidentally, our results indicate that some of the compounds included in the volatile fraction contribute to spasmolytic effect of phytocomplex and that linalool is the most efficacious and potent constituent of BEO to counteract gut contractions elicited by the different spasmogenic agents. Particularly, linalool induces a dose-dependent reduction of the cholinergic contraction in all gut tissues investigated, while limonene and α-pinene elicit a relaxant effect only at the highest concentration used and linalyl acetate shows a concentration-dependent effect only in jejunum and ileum. Moreover, the monoterpene linalool is active at concentrations ten times less than those used with the other compounds investigated. The results obtained by testing BEO constituents show that linalool is also the most effective to counteract the KCl-evoked contraction. Our results support data on the antispasmodic activity of different essential oils from *Citrus* spp on gut smooth muscles reviewed by Heghes and colleagues (2019) [22]. Particularly, essential oil from *Citrus aurantifolia* demonstrated a spasmolytic activity with a progressive reduction in amplitude of contractions and muscle tone in isolated rabbit jejunum preparation [23]. More recently, Sánchez-Recillas and colleagues (2017) reported a spasmolytic activity of the extract obtained from two species of *Citrus sinensis* on rat ileum [24]. Interestingly, several data suggest that the antispasmodic effect of phytocomplexes could be attributed to volatile compounds [22].

The ability of BEO and linalool to interfere with the contractions elicited by different spasmogenic compounds seems to suggest that their effect may be due to the modulation of intracellular processes common to different stimuli rather than a receptor-type mechanism and supports data by Straface and colleagues [21]. Particularly, these authors reported that BEO and linalool showed an inhibitory effect on muscle contraction evoked by a submaximally-effective concentration of KCl in rat and human colon, suggesting the ability to act directly on the smooth muscle cells. Straface and colleagues also suggested that BEO, mainly through linalool, could be able of acting on different targets, highlighting the ability to influence Na^+^, Ca^2+^ and K^+^ intracellular levels [21]. In nerve cells, the increase in the intracellular levels of the Na^+^ ions is fundamental for initiating the genesis of an action potential, which in turn determines an intracellular increase of Ca^2+^ ions to guarantee the fusion of synaptic vesicles on the presynaptic membrane. It follows that a reduction in the intracellular levels of Na^+^ reduces the possibility of generating action potential and consequently release of neurotransmitters, including acetylcholine [25]. Incidentally, Vatanparast and colleagues (2017) reported an inhibitory action of linalool on Na^+^-channels in central neurons of *Caucasotachea atrolabiata* [26]. Moreover, Narusuye and colleagues (2005) reported that linalool showed inhibition of Na^+^ current other than other ions currents in olfactory receptor cells of rat [27]. In mouse aortic rings, BEO may act on K^+^-channels and blocks the membrane voltage-gated Ca^2+^ channels, producing endothelium-independent vasorelaxation given by hyperpolarization and by inhibition of Ca^2+^ influx, fundamental in the interaction between actin and myosin filaments [28]. Although it seems likely that in their ability to inhibit neuromuscular contractions a major activity of phytocomplex is to act directly at the smooth muscle to cause muscle relaxation, it remains a possibility that it might also directly inhibit enteric nerve function. Moreover, even if with limited evidence, it is also possible that BEO, through linalool, via noncompetitive mechanisms may inhibit G protein-coupled receptors, blocking the contractile response that would ensue [29]. Interestingly, high concentrations of linalool also activate in human and mouse TRPM8 receptor, specifically expressed in subpopulations of neurons instrumental in sensing pain and temperature [30].

All these data suggest that antispasmodic activity of bergamot essential oil underlies the involvement of complex mechanisms and deserve further investigation.

Overall, our results confirm that BEO is endowed with reproducible antispasmodic effects, mainly through linalool, and it could be helpful in the complementary treatment of intestinal disorder related to excessive smooth muscle contractility. Particularly, the phytocomplex might be useful in the treatment of irritable bowel syndrome (IBS), a chronic, functional gastrointestinal syndrome characterized by relapsing abdominal pain and altered bowel habits, with either predominant symptoms of diarrhoea and constipation [31]. Currently, a natural remedy, peppermint oil, is widely used in complementary treatment to improve symptoms in patients with IBS [32,33]. Interestingly, peppermint oil gives an antispasmodic action associated with its main component menthol, which blocks the Ca^2+^ influx by actions on the L-type Ca^2+^ channels [34].

IBS is a painful condition in which a role for neurotransmitter dysfunction has been envisaged and like all functional gastrointestinal disorders, it is a disorder of brain–gut interactions [35,36]. Pain, by definition, is the dominant symptom experienced by IBS patients and actually there is no effective cure for pain in IBS [37,38]. Visceral pain is also an emotional experience [39,40] and improving anxiety can lead to reduction of the harmful effects of the pain even when it is still present [39,40,41].

In this context, BEO, endowed with reproducible antispasmodic, analgesic and anxiolytic effects, might be really beneficial in the treatment of IBS. However, more investigation is required for clinical translation of the present data, as it occurred for clinical translation of engineered BEO in pain underlying agitation in dementia [42,43,44].

## 5. Conclusions

These results show that BEO is endowed with reproducible antispasmodic effects on colon, jejunum and ileum of rat and that linalool is the most active constituent. Although the mechanism of action deserves further investigation, the whole essential oil and linalool could be beneficial in the complementary treatment of painful intestinal disorder related to excessive smooth muscle contractility such as IBS.

## Figures and Tables

**Figure 1 pharmaceutics-14-00775-f001:**
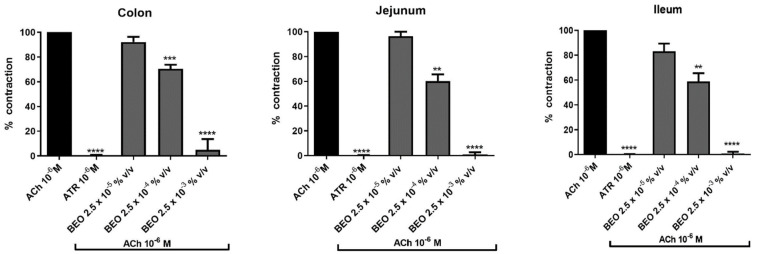
Effect of BEO on ACh-evoked contraction in colon, jejunum and ileum of rat. Each column represents the mean of 3–6 rats. Vertical lines show standard error of mean. ** *p* < 0.01, *** *p* < 0.001 and **** *p* < 0.0001 show the statistical significance between the concentrations of BEO tested on ACh-contraction versus ACh-contraction control (*t*-tests). ACh = acetylcholine, ATR = atropine BEO = bergamot essential oil.

**Figure 2 pharmaceutics-14-00775-f002:**
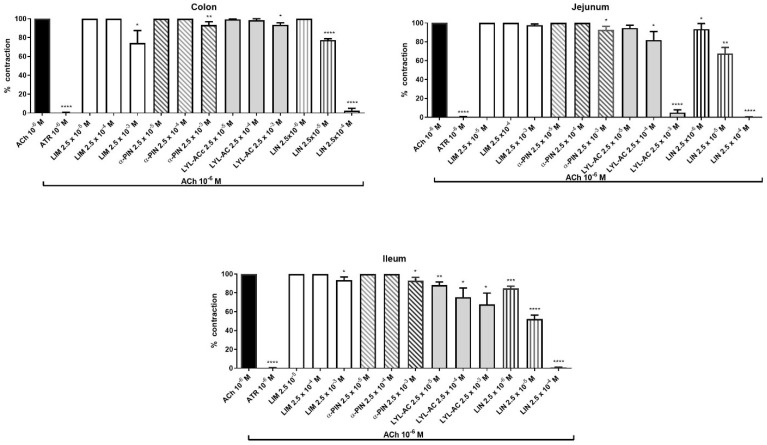
Effect of BEO constituents on ACh-evoked contraction in colon, jejunum and ileum of rat. Each column represents the mean of 3–6 rats. Vertical lines show standard error of mean. * *p* < 0.05, ** *p* < 0.01, *** *p* < 0.001 and **** *p* < 0.0001 show the statistical significance between the concentrations of BEO tested on ACh-contraction versus ACh-contraction control (*t*-tests). ACh = acetylcholine, ATR = atropine, LIM = limonene, α-PIN = α-pinene, LYL-AC = linalyl-acetate, LIN = linalool.

**Figure 3 pharmaceutics-14-00775-f003:**
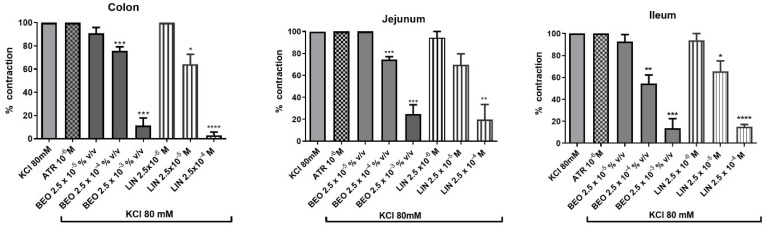
Effect of BEO or linalool on KCl-evoked contraction in colon, jejunum and ileum of rat. Each column represents the mean of 3–6 rats. Vertical lines show standard error of mean. * *p* < 0.05, ** *p* < 0.01, *** *p* < 0.001 and **** *p* < 0.0001 show the statistical significance between the concentrations of BEO tested on ACh-contraction versus ACh-contraction control (*t*-tests). KCl = potassium cloride, ATR = atropine, BEO = bergamot essential oil, LIN = linalool.

## Data Availability

All data available are reported in the article.

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
