# Peer review of "Antispasmodic Effect of Bergamot Essential Oil on Rat Isolated Gut Tissues"

_pharmaceutics, 2022, doi:10.3390/pharmaceutics14040775_

Round 1

Reviewer 1 Report

The manuscript pharmaceutics-1655189 entitled “Antispasmodic effect of bergamot essential oil on rat isolated gut tissues” reports the investigation performed on the impact of the use of crude Bergamot EO and its main components linalool, limonene, linalyl acetate and α-pinene on cholinergically- and non-cholinergically-mediated contractions that were  induced previously by acetylcholine (ACh) and potassium chloride (KCl), respectively on different gut tissues of rat (colon, jejunum and ileum). The authors used an appropriate approach and the results sound scientific. Minor alterations are recommended.

Line 106: check the formula of NaCl and KCl (it is not I).

Line 114: after % add a comma and v/v is into brackets, all over the manuscript.

Line 140: Figure legend Each point? this is not clear. The authors mean data? The same in Figure 2 and 3.

Line 190: what volumes? concentrations?

Line 194: what high volume? The authors tested different concentrations or different volumes? The same in line 203.

Line 220: change to spp. (not in italic)

Author Response

Thanks for your comments and suggestions that will improve the accuracy of the manuscript

  • Line 106: check the formula of NaCl and KCl (it is not I).

The typing error has been correct

  • Line 114: after % add a comma and v/v is into brackets, all over the manuscript.

It was an oversight then, all over the manuscript, a comma has been adding after % and v/v has been put into brackets.

  • Line 140: Figure legend Each point? this is not clear. The authors mean data? The same in Figure 2 and 3.

We agree with the reviewer and we replaced “point” with “column”

  • Line 190: what volumes? concentrations? Line 194: what high volume? The authors tested different concentrations or different volumes? The same in line 203.

Essential oil (BEO) is liquid then its amount is expressed as “volume”; the amount of limonene, α-pinene, linalyl acetate and linalool, purchased from Sigma as a powder, is expressed as concentration (M). In line 198 we changed the term “concentrations” with “volumes” because data referred to essential oil.

  • Line 220: change to spp. (not in italic)

We changed “spp” with “spp (not in italic font)

Reviewer 2 Report

The manuscript is focused on a study of antispasmodic effects of BEO and some of its' active constituents on isolated rat digestive tract segments. The results are interesting, confirming a potential role of bergamot EO and especially linalool as potential solutions in the therapy of digestive spasms. However, a few corrections are needed:

  1. In Introduction, Lines 37-38 it is written that "belonging to the Rutaceae family of the Citrus genus" . Please reformulate the phrase because taxonomically, a Genus is inferior to a Family.
  2. In section 2.2 Animals, the details concerning food and water are repeated twice in Lines 96-97 and 100-101.
  3. In section 2.3 Experimental protocol, the authors should specify that they used atropine as a reference antispasmodic agent in Ach-evoked contractions. Also, the last details concerning organ mounting in the organ bath should be placed ahead of the details concerning substance administration.And finally, perhaps in a future study, the authors could provide additional details on the potential mechanisms of BEO and linalool, e.g. calculating pA2 parameter for BEO for a better understanding of the antagonsim of Ach-induced contractions. It would be interesting to see the magnitude of the effect compared with a known muscarinic anatgonist like atropine, using full concentration-response curves.

Author Response

Thanks for your comments and suggestions that will improve the manuscript

  • In Introduction, Lines 37-38 it is written that "belonging to the Rutaceae family of the Citrus genus". Please reformulate the phrase because taxonomically, a Genus is inferior to a Family.

Accordingly to the reviewer, the sentence has been reformulated as “belonging to genus Citrus, family Rutaceae”

  • In section 2.2 Animals, the details concerning food and water are repeated twice in Lines 96-97 and 100-101.

Accordingly to the reviewer, we deleted sentence in lines 100-101 “Animals were provided with free access to the standard rodent diet and purified drinking water”.

  • In section 2.3 Experimental protocol, the authors should specify that they used atropine as a reference antispasmodic agent in Ach-evoked contractions. Also, the last details concerning organ mounting in the organ bath should be placed ahead of the details concerning substance administration. And finally, perhaps in a future study, the authors could provide additional details on the potential mechanisms of BEO and linalool, e.g. calculating pA2 parameter for BEO for a better understanding of the antagonism of Ach-induced contractions. It would be interesting to see the magnitude of the effect compared with a known muscarinic antagonist like atropine, using full concentration-response curves.

Accordingly to the reviewer, we added in the text that “atropine has been used as a reference antispasmodic agent in Ach-evoked contractions”. In addition, concerning organ mounting in the bath we placed the last phraseThe distal end of each segment was tied to an organ holder and the proximal end was secured with a silk thread to an isometric force transducer (Ugo Basile 7005, Italy). Mechanical activity was digitized on an A/D converter, visualized, recorded and analyzed on a personal computer using the Data Capsule-Evo system (Ugo Basile, Italy)” ahead of the details concerning substance administration.

We appreciate very much your suggestions about calculating pA2 parameter. We are going to plane new experiments to elucidate the mechanisms underlying spasmolytic effect of BEO and pA2 parameter will be evaluated.
